# A Family of Five-Level Pseudo-Totem Pole Dual Boost Converters

Qingsong Zhao [1], Guixi Miao [2], Hong Dai [3,*], Cheng Jing [3], Jianyuan Xu [1], Wenjing Li [3] and Hui Ma [3,*]

1 School of Electrical Engineering, China Shenyang University of Technology, Shenyang 110870, China; hi295@163.com (Q.Z.); intxjy@163.com (J.X.)
2 State Gird Henan Anyang Power Supply Co., Ltd., Anyang 455000, China; miao.gx@163.com
3 College of Electrical Engineering and New Energy, China Three Gorges University, Yichang 443002, China; jc1679977167@163.com (C.J.); liwenjing@ctgu.edu.cn (W.L.)
* Correspondence: daihong@ctgu.edu.cn (H.D.); mahuizz119@126.com (H.M.)

**Abstract:** In this paper, based on the pseudo-totem pole (PTP) circuit, a family of five-level PTP dual boost converters (PDBC) is proposed. A dual boost converter has some unique advantages, such as having no risk of bridge arm shoot-through and no problems related to switch body diode reverse recovery; thus, it has a good potential for applications. First, the derivation process, working principle, modulation and strategy of the topology are analyzed. Further, the number of power devices, switch voltages and current stress of the proposed topology is analyzed. Finally, a representative five-level PDBC experimental prototype is designed with AC input 220 V/50 Hz, DC output 400 V/1 kW, and peak efficiency of 98.27%. The experimental results show that the five-level PDBC proposed in this paper has higher efficiency and the correctness of its topology is verified.

**Keywords:** five-level dual boost converter; pseudo-totem pole (PTP); efficiency; voltage and current stress

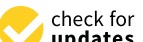



## 1. Introduction

With the development of power electronics technology, the neutral point clamped (NPC) multi-level converter has attracted more and more scholars' attention [1,2]. Multi-level NPC converters have the advantages of high efficiency, high power density, low total harmonic distortion (THD) and low voltage and current stress [3–5]. Therefore, multi-level NPC converters are widely used in medium and high power, medium voltage applications [6–9], such as electric vehicles (EVs), motor drives, electrical railway traction, and DC-bus microgrids [10–13].

Compared to an IGBT, a power MOSFET has some excellent characteristics, such as high frequency operation andlow switching and conduction losses. Therefore, power MOSFETs are widely used in low-power converters to improve efficiency and reduce volume. However, due to the poor reverse recovery characteristics of the body diode, those that are MOSFET-based have a risk of device failure, which is related to the phase-leg shoot, and this may cause false triggering of the gate voltage. Therefore, conventional H-bridge converters often use IGBTs [14–17]. The dual boost/buck converter can avoid the shoot-through problem effectively, and the freewheeling current flows through the independent diode to resolve the MOSFET body diode reverse recovery problem [14–21]. However, due to the two inductance structures of the dual boost/buck converter, the volume and weight of the converter will increase. With this in mind, the multi-level dual boost/buck converter has been proposed in [16,17,22], to reduce the size and weight of this type of topology.

The bridgeless converter has a high power factor (PF), low THD and high efficiency. This type includes the basic bridgeless converters, bidirectional-switch bridgeless converters, totem pole bridgeless converters and PTP bridgeless converters [22–24]. Among these, the PTP bridgeless converter has a simple structure, few semiconductor devices

conducting in the series loop and a natural dual-boost structure. In order to increase the power density and reduce the THD, the PTP bridgeless converter must work at a higher frequency. Moreover, all the active devices in the circuit carry a DC voltage, which causes high voltage stress [3].

Figure 1 shows two common five-level converter topologies. Figure 1a is conventional five-level converter (CFLC), which is widely used in bidirectional power flow scenarios such as vehicle to grid (V2G) technologies. However, the number of switches is large; thus, this can be further optimized to reduce the number of switches. Figure 1b is a unidirectional five-level converter proposed in [25], which uses only one NPC bridge arm in the CFLC to reduce the number of switches. However, there is no topology derivation process. In [4], a new five-level converter with high power density is proposed and the current stress is analyzed; however, the number of active devices is large, and the cost is high. In [14,16], the PTP bridgeless structure is adopted to achieve a five-level, dual-buck, grid-connected inverter, and these papers summarize a topology structure method to expand the circuit. However, the utilization of these two input inductors is low.

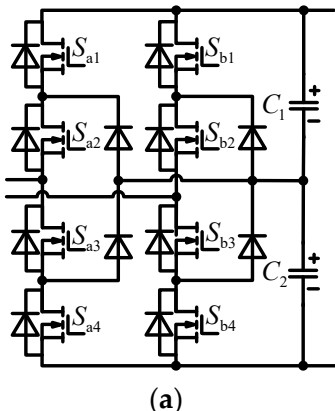
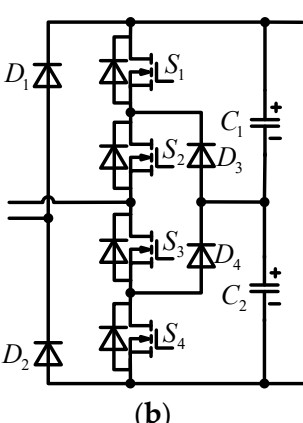

(a)                                                                 (b)

**Figure 1.** The common five-level converter topologies. (**a**) Conventional five-level converter; (**b**) Converter proposed in [25].

Based on the above analysis, the bridge arm of the topology proposed in [25] is used as the prototype to obtain new five-level bridge arms. By cascading the five-level bridge arms with the three-level PTP circuit, a family of five-level PDBCs is proposed. The proposed PDBC optimizes the three-level PTP converter to address the large volume and large switching losses and to retain the dual boost structure; thus, it has high reliability.

The structure of this paper is as follows. Section 2 derives the PDBC in detail and analyzes the working principle. Section 3 designs the control and modulation strategy of PDBC. In Section 4, the voltage and current stress are analyzed. Section 5 conducts experimental verification on PDBC. Finally, Section 6 concludes the paper.

## 2. Topological Derivation and Operating Principle

### 2.1. Topological Derivation

Figure 2 shows the three most commonly used clamping bridge arm structures in five-level converters. A new five-level converter can be obtained by cascading these three clamping bridge arm structures with a three-level circuit.

In this paper, the bridge arm shown in Figure 2b is used to generate a new bridge arm, as shown in Figure 3. For ease of description, the bridge arms are numbered below each bridge arms, i.e., A1, A2. Firstly, the two diodes on the right side of A4 are replaced with MOSFETs to get A1*. Then, $S_2$ and $S_4$ of A1* are removed and reconnected to get a new bridge arm A2, as shown in Figure 3a. In the same way, $S_1$ and $S_3$ of A1* are removed and reconnected to obtain a new bridge arm, A3, as shown in Figure 3b.

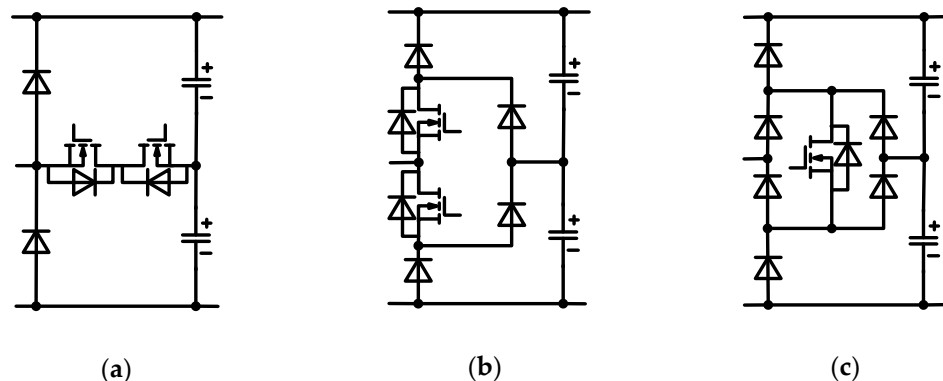

**Figure 2.** Three most commonly used clamping bridge arm structures. (**a**) T-type three-level bridge arm structure; (**b**) Diode Neutral Point Clamping (DNPC) three-level bridge arm structure; (**c**) Single switch three-level bridge arm structure.

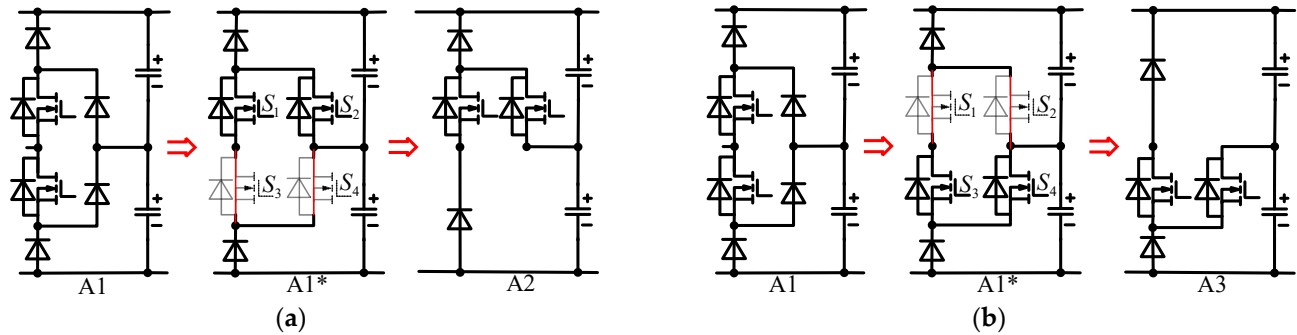

**Figure 3.** Topology derivation. (**a**) Topology derivation of A2; (**b**) Topology derivation of A3.

Through the bridge arm transformation in Figure 3, new bridge arms A2 and A3 are derived. Figure 4 shows a family of five-level PTP dual boost converters, which are named as PDBC-I, PDBC-II and PDBC-III. It can be seen in Figure 4 that the PTP dual boost converter is cascaded with Figure 2a, A2, and A3 to generate the PDBCs.

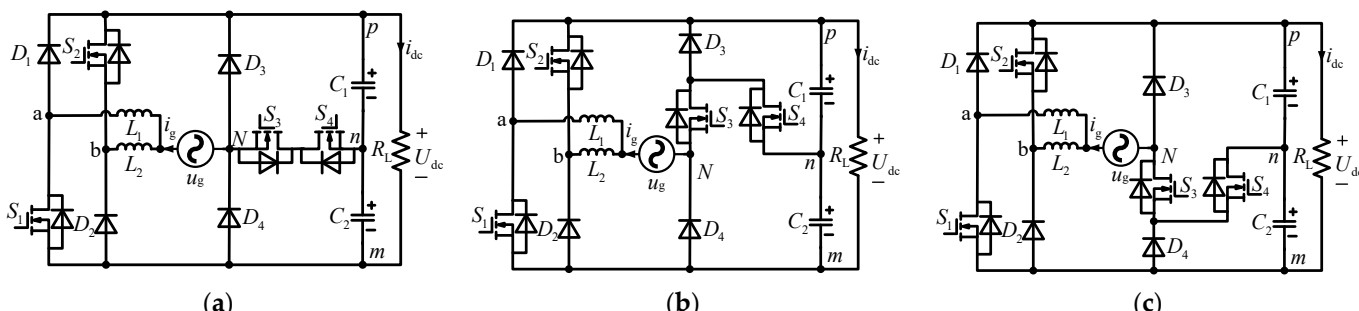

**Figure 4.** A family of five-level PTP dual boost converters. (**a**) PDBC-I; (**b**) PDBC-II; (**c**) PDBC-III.

Compared to the traditional five-level converter, a family of PDBCs proposed in this paper reduces the number of MOSFETS by four, so it can effectively reduce the converter loss. It is worth mentioning that the three power flow converters proposed in this paper can all achieve bidirectional power flow after a slight circuit modification.

### 2.2. Operating Principle

Based on the above analysis, this paper uses the PDBC-II as an example for the working mode analysis. Figure 5 shows the key waveforms of one power cycle, which are corresponding to the six working modes in Figure 6. For ease of analysis, it is assumed that the circuit works in continuous conduction mode (CCM), the inductances and capacitors

are large enough, the capacitor voltage $U_{C1} = U_{C2} = U_{dc}/2$ and the DC voltage, $U_{dc}$, remains constant. Figure 6 shows the six working modes of PDBC-II, which correspond to 0, $\pm 0.5 U_{dc}$ and $\pm U_{dc}$. The six working modes are analyzed as follows.

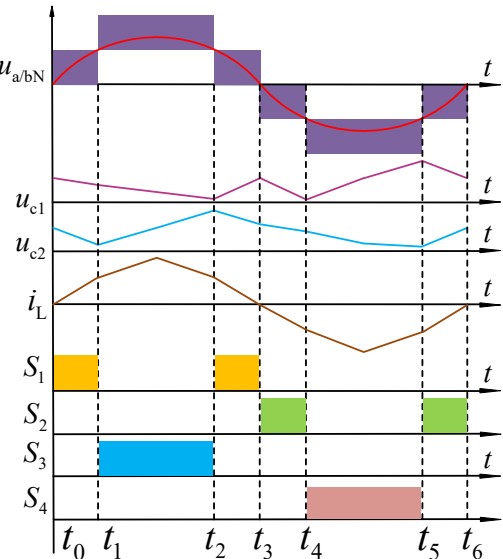

**Figure 5.** Key waveforms of six working modes in one cycle.

In Working mode 1, during [$t_0$, $t_1$] shown in Figure 5, the grid voltages are $u_g > 0$, $u_{aN} = 0$ and $u_{bN} = u_g$. In this mode, the current path is shown in Figure 6a. As shown in Figure 6a, the switch $S_1$ is ON, the diode $D_4$ is ON and the other active components are OFF. The capacitors $C_1$ and $C_2$ are discharging to the load $R_L$.

In Working mode 2, during [$t_1$, $t_2$] shown in Figure 5, the grid voltages are $u_g > 0$ and $u_{aN} = u_{bN} = U_{dc}/2$. In this mode, the current path is shown in Figure 6b. As shown in Figure 6b, the switch $S_3$ is ON, the body diodes of $S_2$ and $S_4$ are ON, the diode $D_1$ is ON and the other active components are OFF. The capacitor $C_1$ is charging and $C_2$ is discharging to the load $R_L$.

In Working mode 3, during [$t_2$, $t_3$] shown in Figure 5, the grid voltages are $u_g > 0$ and $u_{aN} = u_{bN} = U_{dc}$. In this mode, the current path is shown in Figure 6c. As shown in Figure 6c, the body diode of $S_2$ is ON, the diodes $D_1$ and $D_4$ are ON and the other active components are OFF. The capacitors $C_1$ and $C_2$ are charging, and the AC power supplies to the load $R_L$.

In Working mode 4, during [$t_3$, $t_4$] shown in Figure 5, the grid voltages are $u_g < 0$, $u_{bN} = u_g$ and $u_{bN} = 0$. In this mode, the current path is shown in Figure 6d. As shown in Figure 6d, the switch $S_2$ is ON, the body diode of $S_3$ is ON, the diode $D_3$ is ON and the other active components are OFF. The capacitors $C_1$ and $C_2$ are discharging to the load $R_L$.

In Working mode 5, during [$t_4$, $t_5$] shown in Figure 5, the grid voltage $u_g < 0$ and $u_{aN} = u_{bN} = -U_{dc}/2$. In this mode, the current path is shown in Figure 6e. As shown in Figure 6e, the switch $S_4$ is ON, the body diodes of $S_1$ and $S_3$ are ON, the diode $D_2$ is ON and the other active components are OFF. The capacitor $C_2$ is charging and $C_1$ is discharging to the load $R_L$.

In Working mode 6, during [$t_5$, $t_6$] shown in Figure 5, the grid voltages are $u_g < 0$ and $u_{aN} = u_{bN} = -U_{dc}$. In this mode, the current path is shown in Figure 6f. As shown in Figure 6f, the body diodes of $S_1$ and $S_3$ are ON, the diodes $D_2$ and $D_3$ are ON, and the other active components are OFF. The capacitors $C_1$ and $C_2$ are charging, and the AC power supplies to the load $R_L$.

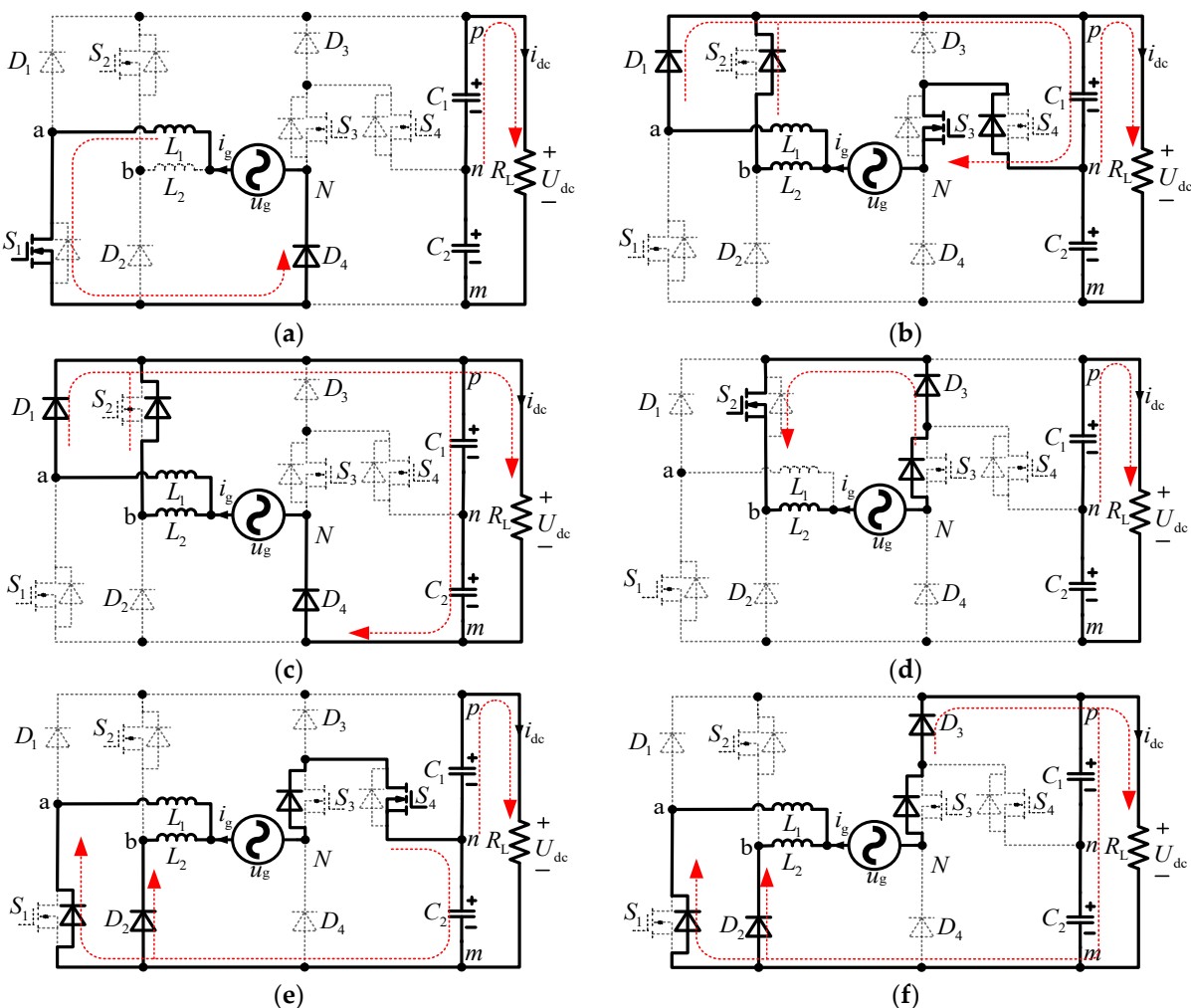

**Figure 6.** The six working mode of PDBC-II. (**a**) Working mode 1; (**b**) Working mode 2; (**c**) Working mode 3; (**d**) Working mode 4; (**e**) Working mode 5; (**f**) Working mode 6.

Based on the six working mode analyses above, the switching pulse distribution of the PDBC-II is summarized in Table 1. Where "0" and "1" represent switching off and on, "↑" and "↓" represent capacitor charging and discharging, respectively. It can be seen from Table 1 that the PDBC-II has at most one switch ON in each mode, so the PDBC-II has a lower conduction loss.

**Table 1.** Switching pulse distribution of PDBC-II.

| Mode | $i_g$ | $S_1$ | $S_2$ | $S_3$ | $S_4$ | $C_1$ | $C_2$ | $u_{aN}$ | $u_{bN}$ |
|------|-------|-------|-------|-------|-------|-------|-------|----------|----------|
| 1 | >0 | 1 | 0 | 0 | 0 | ↓ | ↓ | 0 | $u_g$ |
| 2 | >0 | 0 | 0 | 1 | 0 | ↑ | ↓ | $U_{dc}/2$ | $U_{dc}/2$ |
| 3 | >0 | 0 | 0 | 0 | 0 | ↑ | ↑ | $U_{dc}$ | $U_{dc}$ |
| 4 | <0 | 0 | 1 | 0 | 0 | ↓ | ↓ | $u_g$ | 0 |
| 5 | <0 | 0 | 0 | 0 | 1 | ↓ | ↑ | $-U_{dc}/2$ | $-U_{dc}/2$ |
| 6 | <0 | 0 | 0 | 0 | 0 | ↑ | ↑ | $-U_{dc}$ | $-U_{dc}$ |

## 3. Control and Modulation Strategy

### 3.1. Control Strategy

In this paper, a double closed-loop control system suited for PDBCs is designed; both the voltage outer loop and the current inner loop adopt the PI controller. The control block

diagram of the PDBCs is shown in Figure 7. The transfer function of the PI link in Figure 7 can be expressed as Equation (1),

$$G_{\mathrm{pi}} = (\mathrm{k_p} \cdot s + \mathrm{k_i})/s, \tag{1}$$

where $\mathrm{k_p}$ and $\mathrm{k_i}$ are the proportional and integral coefficients, respectively.

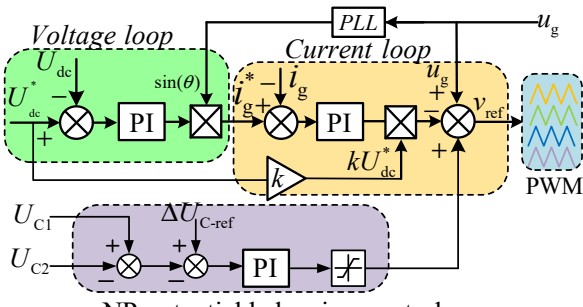

**Figure 7.** Control block diagram of the PDBCs.

The voltage outer loop is used to keep the output voltage $U_{\mathrm{dc}}$ stable; this output is the reference value of the current inner loop. Firstly, the difference between the DC side voltage, $U_{\mathrm{dc}}$, and its reference value, $U_{\mathrm{dc}}$*, is sent into the PI controller to obtain an error signal. Then, the error signal is multiplied by the phase information of the grid voltage $u_{\mathrm{g}}$. As a result, the input reference signal, $i_{\mathrm{g}}$*, of the current inner loop is obtained.

The current inner loop is used to ensure that the waveform of $i_{\mathrm{g}}$ is sinusoidal and controls the power factor close to 1. Firstly, the difference between the AC side current, $i_{\mathrm{g}}$, and the reference signal, $i_{\mathrm{g}}$*, (the output of the voltage outer loop) is sent to the PI controller. Then, the output of the PI controller is multiplied by $kU_{\mathrm{dc}}$*. Finally, its output is the difference between this product and the grid voltage, $u_{\mathrm{g}}$, to get the PWM reference signal, $v_{\mathrm{ref}}$.

The neutral point (NP) balance of the DC side capacitor is attained by the phase delay control method in this paper. Firstly, the difference between the voltage signals of the capacitors $C_1$ and $C_2$ is compared to the reference value. Then, the result is sent to the PI controller to obtain the correction signal, which is added to the modulation wave after limiting the amplitude. Therefore, the NP voltage balance is achieved by adjusting the capacitor charging and discharging time.

*3.2. Modulation Strategy*

In this paper, the multi-carrier PWM modulation strategy is used to generate a switching pulse distribution for the proposed PDBCs. The PDBC-II is used as an example to illustrate the switching pulse distribution and bridge arm voltages $u_{\mathrm{aN}}$, $u_{\mathrm{bN}}$ of the multi-carrier PWM modulation strategy, as shown in Figure 8, where $v_{\mathrm{c1}}(t)$, $v_{\mathrm{c2}}(t)$, $v_{\mathrm{c3}}(t)$, $v_{\mathrm{c4}}(t)$ and $v_{\mathrm{ref}}(t)$ represent four carrier signals and the reference signal, respectively. The reference signal, $v_{\mathrm{ref}}(t)$, is compared to $v_{\mathrm{c1}}(t)$, $v_{\mathrm{c2}}(t)$, $v_{\mathrm{c3}}(t)$ and $v_{\mathrm{c4}}(t)$ to obtain the switching pulse waveform of the switches $S_1$, $S_2$, $S_3$ and $S_4$, respectively.

In the positive half-cycle of the PWM modulation strategy, the duty cycle can be derived from Equations (2)–(6), below. For these Equations, $U_{\mathrm{g,max}}$, $T_{\mathrm{on}}$ and $T_{\mathrm{off}}$ are the amplitude of AC voltage and the turn on and off times of the switch, respectively. The modulation ratio M and duty ratio $D$ can be defined as follows:

$$M = U_{\mathrm{g,max}}/U_{\mathrm{dc}} \tag{2}$$

and

$$D = T_{\mathrm{on}}/(T_{\mathrm{on}} + T_{\mathrm{off}}). \tag{3}$$

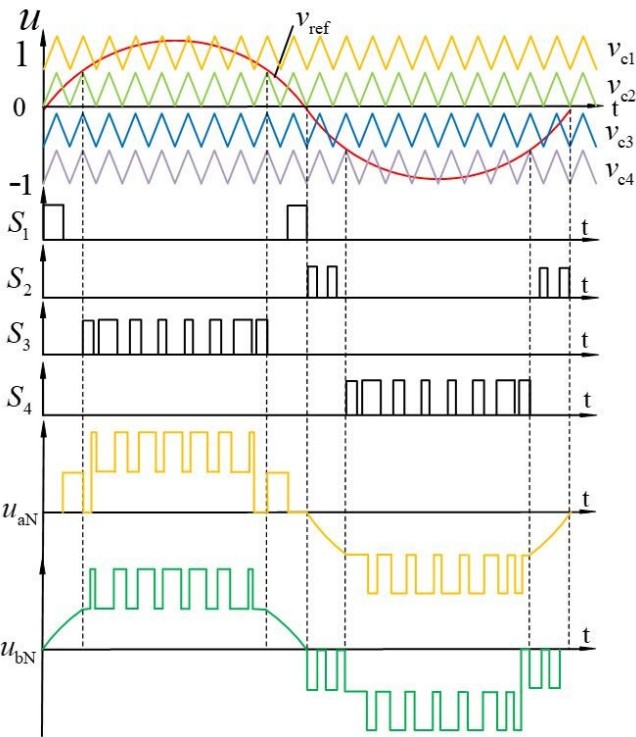

**Figure 8.** Diagram of modulation strategy.

When $0 < v_{ref}(t) < 0.5$, the PDBCs operate alternately in Working modes 1 and 2. At this time, the inductance volt-second balance has the following relationship:

$$\left(u_g - 0.5U_{dc}\right) \cdot T_{on} = \left(U_{dc} - u_g\right) \cdot T_{off}. \tag{4}$$

The duty cycle $D_1$ is derived from Equations (2)–(4):

$$D_1 = 1 - 2M\sin(wt). \tag{5}$$

Similarly, when $0.5 < v_{ref}(t) < 1$, the PDBCs operate alternately in Working modes 2 and 3. The duty cycle $D_2$ can be obtained as follows:

$$D_2 = 2 - 2M\sin(wt). \tag{6}$$

When the PDBCs operate in the positive half-cycle, the resulting duty cycle change curve, based on Equations (5) and (6), is shown in Figure 9, where the green curve represents the duty cycle $D_1$, the yellow curve represents the duty cycle $D_2$ and $\theta_1$ and $\theta_2$ are the angles of alternation between $v_{c1}(t)$ and $v_{c2}(t)$, respectively.

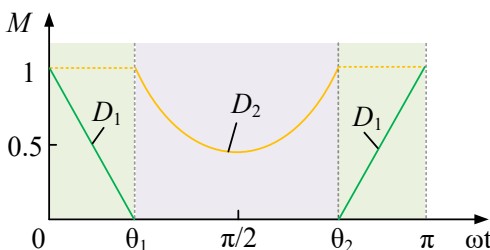

**Figure 9.** Change curve of duty cycle in the positive half-cycle.

It is assumed that, the amplitude of $v_{\text{ref}}(t)$ is 1 (after normalization) and the $\theta_1$ and $\theta_2$ can be calculated as follows:

$$\theta_1 = \arcsin 0.5 = \frac{\pi}{6} \tag{7}$$

$$D_2 = 2 - 2M \sin(wt). \tag{8}$$

## 4. Performance Analysis

### 4.1. Voltage Stress Analysis

Based on the working mode analysis of Figure 6, the switch and diode voltage stresses of the three PDBCs are summarized, as shown in Table 2. It can be seen from Table 2 that part of the components voltage stress on the three-level bridge is reduced by half. In addition, the PDBC-II and PDBC-III have three components with voltage stresses of reduced by half, while the PDBC-I has only two components with voltage stresses reduced by half. Therefore, in terms of cost, PDBC-II and PDBC-III have lower costs compared to PDBC-I.

**Table 2.** Voltage stress of three PDBCs.

| Components | PDBC-I | PDBC-II | PDBC-III |
|:---:|:---:|:---:|:---:|
| $S_{1,2}$ | $U_{\text{dc}}$ | $U_{\text{dc}}$ | $U_{\text{dc}}$ |
| $S_{3,4}$ | $U_{\text{dc}}/2$ | $U_{\text{dc}}/2$ | $U_{\text{dc}}/2$ |
| $D_{1,2}$ | $U_{\text{dc}}$ | $U_{\text{dc}}$ | $U_{\text{dc}}$ |
| $D_3$ | $U_{\text{dc}}$ | $U_{\text{dc}}/2$ | $U_{\text{dc}}$ |
| $D_4$ | $U_{\text{dc}}$ | $U_{\text{dc}}$ | $U_{\text{dc}}/2$ |

### 4.2. Current Stress Analysis

To estimate the conduction loss of the switches and diodes, it is necessary to calculate the average and the root mean square (RMS) of the current flowing through the switches and diodes. In this section, the PDBC-II is used as the example in which the current stress analysis is conducted. Figure 10 shows the current stress simulation waveform of PDBC-II in the Matlab/Simulink environment. It is clear, from Figures 9 and 10, that the switching times of the duty cycles $D_1$ and $D_2$ are $T_g/12$ and $5T_g/12$ ($T_g$ is one power frequency cycle), respectively. Based on Figure 10, the theoretical derivation of the average value and RMS value of the current flowing through the semiconductor device can be calculated using Equation (9) through (18). For the following calculations, it is assumed that the AC side current is a sine wave, the DC side voltage remains unchanged, and the switching frequency $f_s$ is much greater than the grid frequency $f_g$.

The current RMS and average of the diodes $D_1$ and $D_2$ are calculated using Equation (9):

$$\begin{cases} I_{D1,2,\text{rms}}^2 = \frac{2}{T_g}\left[\int_0^{\frac{T_g}{12}}(1-D_1)\cdot i_g{}^2 dt + \int_{\frac{T_g}{12}}^{\frac{T_g}{4}}\left(\frac{i_g+0.5I_{g,\max}}{2}\right)^2 dt\right] \\ I_{D1,2,\text{avg}} = \frac{2}{T_g}\left[\int_0^{\frac{T_g}{12}}(1-D_1)\cdot i_g dt + \int_{\frac{T_g}{12}}^{\frac{T_g}{4}}\left(\frac{i_g+0.5I_{g,\max}}{2}\right) dt\right] \end{cases}. \tag{9}$$

Substituting Equation (5) into Equation (9), the current RMS and average of diodes $D_1$ and $D_2$ can be obtained:

$$\begin{cases} I_{D1,2,\text{rms}} = \frac{\sqrt{6}I_{g,\max}}{24\sqrt{\pi}}\cdot\sqrt{6\pi + 15\sqrt{3} + \left(128 - 72\sqrt{3}\right)M} \\ I_{D1,2,\text{avg}} = \frac{I_{g,\max}}{12\pi}\cdot\left[\pi + 3\sqrt{3} + \left(2\pi - 3\sqrt{3}\right)M\right] \end{cases}. \tag{10}$$

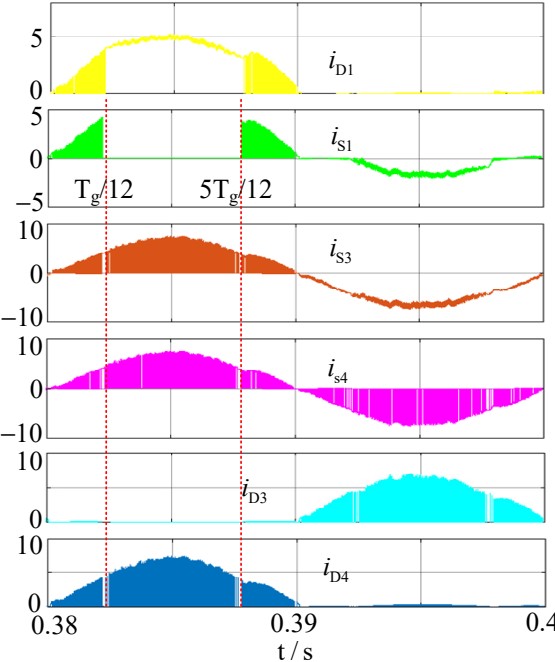

**Figure 10.** The current stress of PDBC-II.

The current RMS and average of the MOSFET $S_1$ and $S_2$ are calculated using Equation (11):

$$
\begin{cases}
I_{S1,2,\text{rms}}^2 = \frac{2}{T_g}\left[ \int_0^{\frac{T_g}{12}} D_1 \cdot i_g{}^2 dt + \int_{\frac{7T_g}{12}}^{\frac{3T_g}{4}} \left( \frac{i_g + 0.5I_{g,\text{max}}}{2} \right)^2 dt \right] \\
I_{S1,2,avg} = \frac{2}{T_g}\left[ \int_0^{\frac{T_g}{12}} D_1 \cdot i_g dt + \int_{\frac{7T_g}{12}}^{\frac{3T_g}{4}} \left| \frac{i_g + I_{g,\text{max}}}{2} \right| dt \right]
\end{cases}
. \tag{11}
$$

Substituting Equation (5) into Equation (11), the current RMS and average of the MOSFET $S_1$ and $S_2$ can be obtained:

$$
\begin{cases}
I_{S1,2,\text{rms}} = \frac{\sqrt{6}I_{g,\text{max}}}{24\sqrt{\pi}} \cdot \sqrt{14\pi - 21\sqrt{3} + \left(72\sqrt{3} - 128\right)M} \\
I_{S1,2avg} = \frac{I_{g,\text{max}}}{12\pi} \cdot \left[ 12 - \pi - 3\sqrt{3} + \left(3\sqrt{3} - 2\pi\right)M \right]
\end{cases}
. \tag{12}
$$

The current RMS and average of MOSFET $S_3$ is calculated using Equation (13):

$$
\begin{cases}
I_{S3,\text{rms}}^2 = \frac{4}{T_g}\left[ \int_0^{\frac{T_g}{12}} D_1 \cdot i_g{}^2 dt + \int_{\frac{T_g}{12}}^{\frac{T_g}{4}} D_2 \cdot i_g{}^2 dt + \int_{\frac{T_g}{2}}^{\frac{3T_g}{4}} i_g{}^2 dt \right] \\
I_{S3,avg} = \frac{2}{T_g}\left[ \int_0^{\frac{T_g}{12}} D_1 \cdot i_g dt + \int_{\frac{T_g}{12}}^{\frac{T_g}{4}} D_2 \cdot i_g dt + \int_{\frac{T_g}{2}}^{\frac{3T_g}{4}} \left| i_g \right| dt \right]
\end{cases}
. \tag{13}
$$

Substituting Equations (5) and (6) into Equation (13), the current RMS and average of MOSFET $S_3$ can be obtained:

$$
\begin{cases}
I_{S3,\text{rms}} = \frac{I_{g,\text{max}}}{2\sqrt{3}\pi} \cdot \sqrt{3\sqrt{3} + 10\pi - 32M} \\
I_{S3,avg} = \frac{I_{g,\text{max}}}{\pi} \cdot \left( \sqrt{3} + 2 - \pi M \right)
\end{cases}
. \tag{14}
$$

The current RMS and average of MOSFET $S_4$ is calculated using Equation (15):

$$
\begin{cases}
I^2_{S4,rms} = \frac{4}{T_g}\left[\int_0^{\frac{T_g}{12}} D_1 \cdot i_g{}^2 dt + \int_{\frac{T_g}{12}}^{\frac{T_g}{4}} D_2 \cdot i_g{}^2 dt\right] \\
I_{S4,avg} = \frac{4}{T_g}\left[\int_0^{\frac{T_g}{12}} D_1 \cdot i_g dt + \int_{\frac{T_g}{12}}^{\frac{T_g}{4}} D_2 \cdot i_g dt\right]
\end{cases}
. \tag{15}
$$

Substituting Equations (5) and (6) into Equation (15), the current RMS and average of MOSFET $S_4$ can be obtained:

$$
\begin{cases}
I_{S4,rms} = \frac{I_{g,max}}{2\sqrt{3}\pi} \cdot \sqrt{3\sqrt{3} + 10\pi - 32M} \\
I_{S4,avg} = \frac{I_{g,max}}{\pi} \cdot \left(\sqrt{3} + 2 - \pi M\right)
\end{cases}
. \tag{16}
$$

The current RMS and average of diodes $D_3$ and $D_4$ current is calculated using Equation (17):

$$
\begin{cases}
I^2_{D3,4,rms} = \frac{2}{T_g}\left[\int_0^{\frac{T_g}{12}} (1 - D_1) \cdot i_g{}^2 dt + \int_{\frac{T_g}{12}}^{\frac{T_g}{4}} (1 - D_2) \cdot i_g{}^2 dt\right] \\
I_{D3,4,avg} = \frac{4}{T_g}\left[\int_0^{\frac{T_g}{12}} (1 - D_1) \cdot i_g dt + \int_{\frac{T_g}{12}}^{\frac{T_g}{4}} (1 - D_2) \cdot i_g dt\right]
\end{cases}
. \tag{17}
$$

Substituting Equations (5) and (6) into Equation (17), the current RMS and average of diodes $D_3$ and $D_4$ can be obtained:

$$
\begin{cases}
I_{D3,4,rms} = \frac{I_{g,max}}{2\sqrt{6}\pi} \cdot \sqrt{32M - 3\sqrt{3} - 4\pi} \\
I_{D3,4,avg} = \frac{I_{g,max}}{2\pi} \cdot \left(\pi M - \sqrt{3}\right)
\end{cases}
. \tag{18}
$$

Figure 11 shows the change curve of the current average and RMS of the switches and diodes with the modulation ratio, *M*, after normalization of $I_{g,max}$. It can be seen from Figure 11 that the current RMS value and average for each of the components have the same trend. Since $I_{g,max}$ are normalized, the trend shown in Figure 11 is also applicable under different power levels, which is of great significance for studying the working principle of the rectifier.

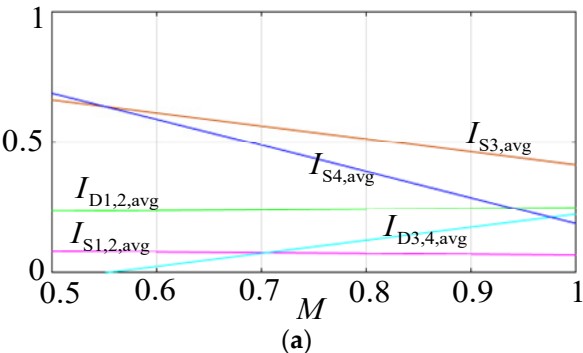 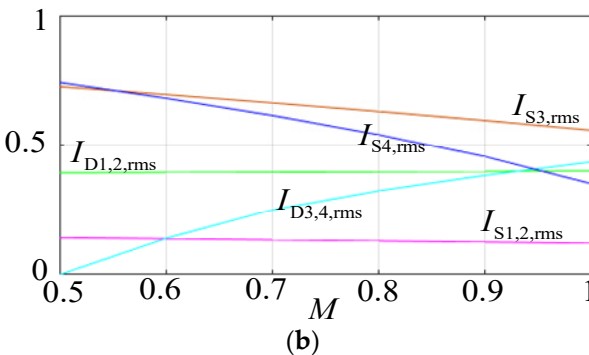

**Figure 11.** Power component average & RMS current stress as functions of the modulation ratio M normalized: $I_{g,max}$. (**a**) Average value; (**b**) RMS value.

To verify the correctness of the current stress formula developed in this paper, the current stress value of the switches and diodes is measured in the Matlab/Simulink environment under a 1 kW power rate. Figure 12 shows the comparison between the calculated value and the simulated value of the current stress. It can be seen from Figure 12, the calculated current stress in this paper is consistent with the simulated value, which verifies the correctness of the current stress formula.

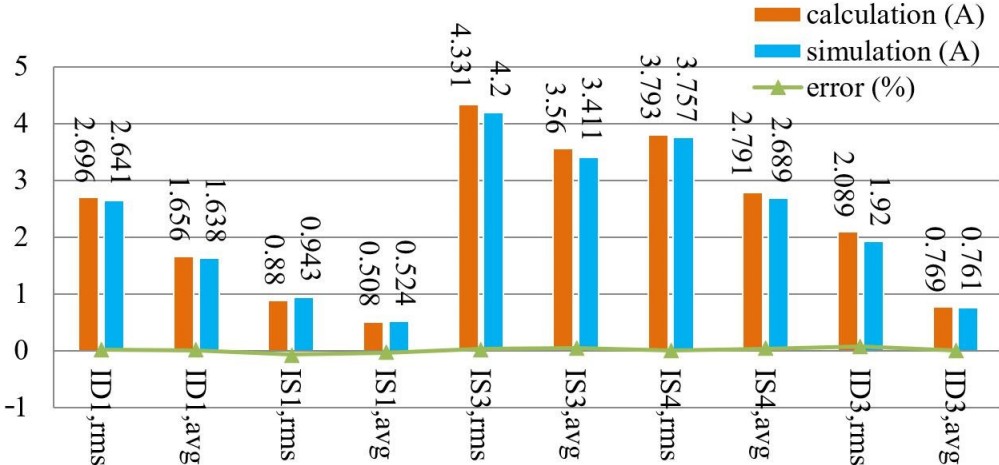

**Figure 12.** Current stress comparison of calculated and simulated value.

### 4.3. Loss Analysis

Based on the above current stress analysis, this section analyzes the loss and efficiency of the proposed PDBCs. The losses of the MOSFET mainly include conduction losses and switching losses. When the MOSFET is on-state, the on-state resistance, $r_{ds}$, results in the conduction losses. The conduction losses $P_{con}$ can be calculated as (19), where $I_{S,rms}$ is the RMS value of the current flowing through the MOSFET, which can be obtained from the current stress analysis:

$$P_{con} = I_{S,rms}^2 \cdot r_{ds}. \tag{19}$$

During the drain current and voltage conversion processes of the MOSFET, the switching losses are generated. These can be calculated using Equation (20), where $U_{in}$, $I_o$, $t_{on}$, $t_{off}$, and $f_s$ are the RMS value of the input voltage, the output current, the switching process drain current and voltage conversion crossover time and the switching frequency, respectively:

$$P_{sw} = 0.5 U_{in} \cdot I_o \cdot \left( t_{on} + t_{off} \right) f_s. \tag{20}$$

Similarly, the power losses of the diode in a cycle consists mainly of static losses and dynamic losses. For the static losses, since the fast recovery diode is employed in this paper, there are only on-state losses considered. The dynamic losses of the diode can be calculated with Equation (21), where $U_f$ is the forward conduction voltage, $I_{D,avg}$ is the average value of the diode:

$$P_{Son} = \int_0^{T_g} u \cdot i \, dt / T_g = U_f \cdot I_{D,avg}. \tag{21}$$

Compared to the reverse recovery time, $t_r$, the forward recovery time, $t_f$, is negligible; therefore, the turn-on losses can be ignored. The turn-off losses can be calculated using Equation (22), where $U_{rp}$ and $I_{rp}$ are the reverse peak voltage and current, $t_b$ is the reverse current fall time:

$$P_{Doff} = \int_{t_0}^{t_2} u_F(t) \cdot i_F(t) \, dt / T_S = 0.5 U_{rp} I_{rp} t_b f_s \tag{22}$$

In this paper, the MOSFETs and diodes employ IRFP450 and RHRP3060, respectively. According to the data sheets, the parameters for IRFP450 and RHRP3060 are shown in Table 3. The switching frequency is $f_s$ = 20 kHz.

**Table 3.** Component parameters.

| Components | Parameters | Values |
|---|---|---|
| IRFP450 | On-state resistance, $r_{ds}$ | 0.4 Ω |
| | On-delay time, $t_{d(on)}$ | 17 ns |
| | Rise time, $t_r$ | 47 ns |
| | Turn-off delay time, $t_{d(off)}$ | 92 ns |
| | Fall time, $t_f$ | 44 ns |
| RHRP3060 | Conduction voltage, $U_f$ | 1.7 V |
| | Reverse peak voltage, $U_{rp}$ | 600 V |
| | Reverse peak current, $I_{rp}$ | 250 μA |
| | Reverse current fall time, $t_b$ | 18 ns |

Based on Equation (9) through (22), the losses of the proposed three PDBCs and the CFLC at different output powers are calculated. According to the results of the loss calculations, the efficiency diagrams at different output power rates are generated, as shown in Figure 13. From this, it can be seen that the four circuits reach peak efficiency at approximately 300 W. However, when the output power is over 300 W, the efficiency declines slowly as the output power increases. The peak efficiency of the PDBC-I is 98.27%, which is the highest in the four circuits. In addition, as the output power level increases, the efficiency of the PDBC-I reaches the highest of the four circuits. Therefore, the PDBC-I is more suitable at a higher power rate. Figure 13b shows the experimental efficiency and theoretical efficiency of the PDBC-II circuit, and the efficiency analysis is expressed in the Figure 13b. The maximum experimental efficiency is approx. 96.5%, which is lower than the theoretical efficiency. Since there are actual circuits that exist as auxiliary circuits (drive circuit, sampling circuit, et al.), the actual loss is higher than those in the theoretical analysis results. Nevertheless, in the theoretical calculation and practical experimental results, the trend of overall efficiency distribution is consistent.

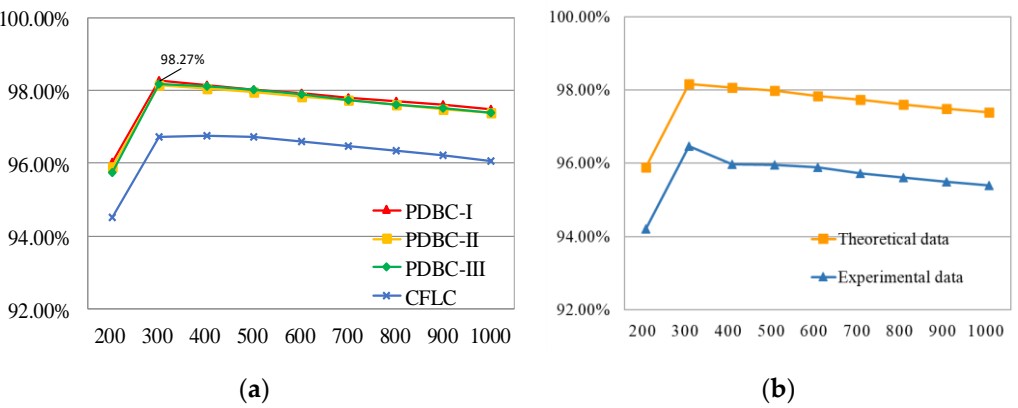

**(a)**                                                        **(b)**

**Figure 13.** Diagram of efficiency analysis. (**a**) Efficiency analysis of the theoretical calculation; (**b**) Experimental efficiency analysis of the PDBC-II.

## 5. Experimental Verification

In order to verify the feasibility of the PDBC proposed in this paper and the correctness of the theoretical analysis, PDBC-II [shown in Figure 4b] is selected for corresponding experimental verification, as shown in Figure 14. The experimental circuit design includes an EMI circuit, auxiliary power supply, main power circuit, signal sampling circuit, MOSFET drive circuit and protection control circuit. The controller employs DSP28335 and the MOSFETs and diodes employ IRFP450 and RHRP3060, respectively; the experimental circuit parameters are shown in Table 4.

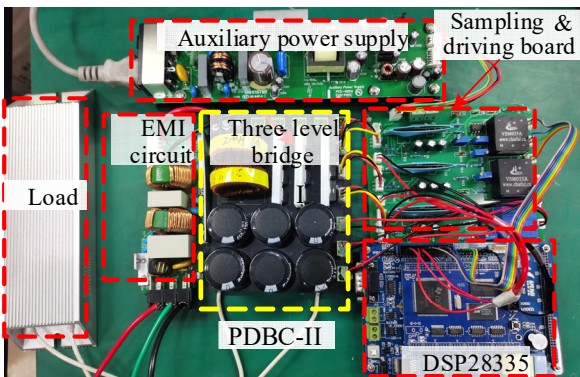

**Figure 14.** Experimental platforms.

**Table 4.** Main parameters.

| Parameters | Label | Value |
|---|---|---|
| Input filter inductors | $L_1$, $L_2$ | 2 mH |
| DC-side capacitors | $C_1$, $C_2$ | 1000 μF |
| Input voltage | $u_g$ | RMS 220 V |
| Output voltage | $U_{dc}$ | 400 V |
| Rated output power | $P_o$ | 1000 W |
| Grid frequency | $f_g$ | 50 Hz |
| Switching frequency | $f_s$ | 20 kHz |

Figure 15 shows the PDBC-II steady-state experimental waveforms of the input and output sides at 1 kW rated power. Figure 15a shows the voltage waveforms of the single-phase AC power supply voltage, $u_g$, the input side current, $i_g$, the DC side voltage, $U_{dc}$, and $U_{C1}$ and $U_{C2}$. As seen from Figure 15a, the input power supply voltage, $u_g$, is in phase with the input side current, $i_g$, and it is sufficiently consistent to achieve power factor correction. The DC side voltage, $U_{dc}$, is stable at 400 V, and the capacitor voltages $U_{C1}$ and $U_{C2}$ remain dynamically balanced at 200 V. Figure 15b shows the bridge arm voltage $u_{aN}$, $u_{bN}$ waveform and the input inductor current $i_{L1}$, $i_{L2}$ waveform. As can be seen from Figure 15b, the waveform of inductor current $i_{L1}$ phase is ahead 180° of the inductor current $i_{L2}$. Similarly, the bridge arm voltage $u_{aN}$, $u_{bN}$ waveform changes in positive and negative half-cycles alternately, which is consistent with Figure 6 and Table 1.

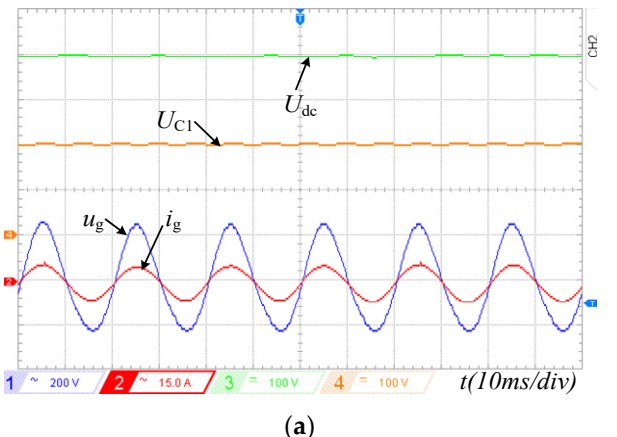

(**a**)

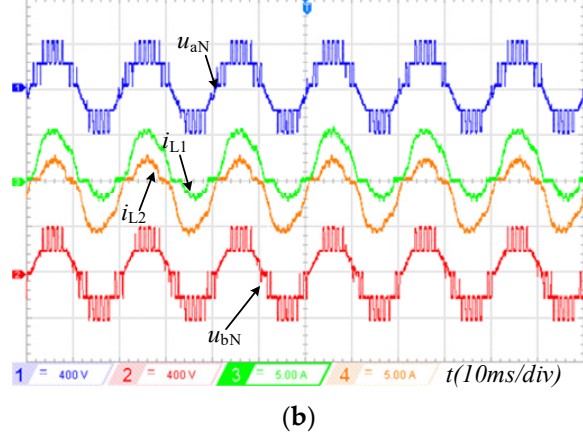

(**b**)

**Figure 15.** Steady-state experimental waveforms. (**a**) The voltage waveforms of power supply voltage $u_g$, current $i_g$, and the DC side voltage $U_{dc}$, $U_{C1}$, $U_{C2}$; (**b**) The bridge arm voltage $u_{aN}$, $u_{bN}$ and the inductor current $i_{L1}$, $i_{L2}$.

Figure 16 shows the PDBC-II switching pulse distribution experimental waveform. As can be seen from Figure 16, the switches $S_1$–$S_4$ have a short, high-frequency switching time. As a result, the switching losses of PDBC-II are relatively low. In addition, the switches $S_3$ and $S_4$ symmetrically action in positive and negative half-cycles, which is consistent with the switching pulse distribution of Table 1.

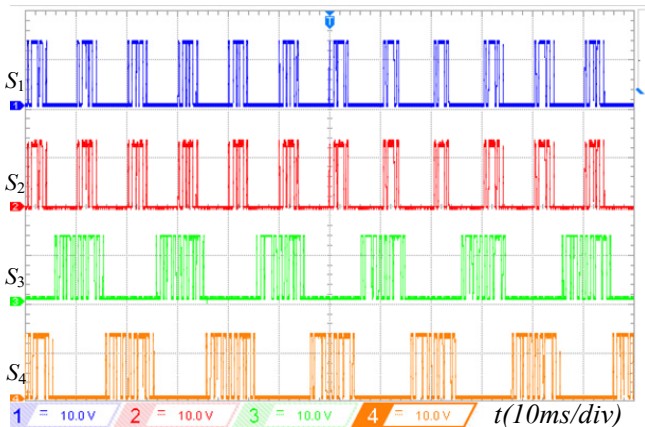

**Figure 16.** Switching pulse distribution experimental waveform.

Figure 17 shows the PDBC-II voltage and current stress experimental waveforms of the switches at 1 kW rated power. Figure 17a shows the voltage stress waveforms of the switches $S_1$–$S_4$. It can be seen from Figure 17a that the maximum voltage stress of the four switches is $U_{S1} = U_{S2} \approx 400$ V and $U_{S3} = U_{S4} \approx 200$ V; i.e., the partial voltage stress is halved, which is consistent with the voltage stress analysis in Table 2. Figure 17b shows the waveform of the inductor current, diode $D_1$ current, $i_{D1}$ and the switch $S_1$ current, $i_{S1}$. As can be seen here, the waveform of the switch current $i_{S1}$ is enveloped by the waveform of the current $i_{L1}$ of the inductor $L_1$ in the positive half cycle. In addition, the currents $i_{D1}$ and $i_{S1}$ change at high frequencies during the $[0, T_g/12]$ and $[5T_g/12, T_g/2]$ (where $T_g$ is the grid period), which is the same as the simulation waveforms in Figure 10.

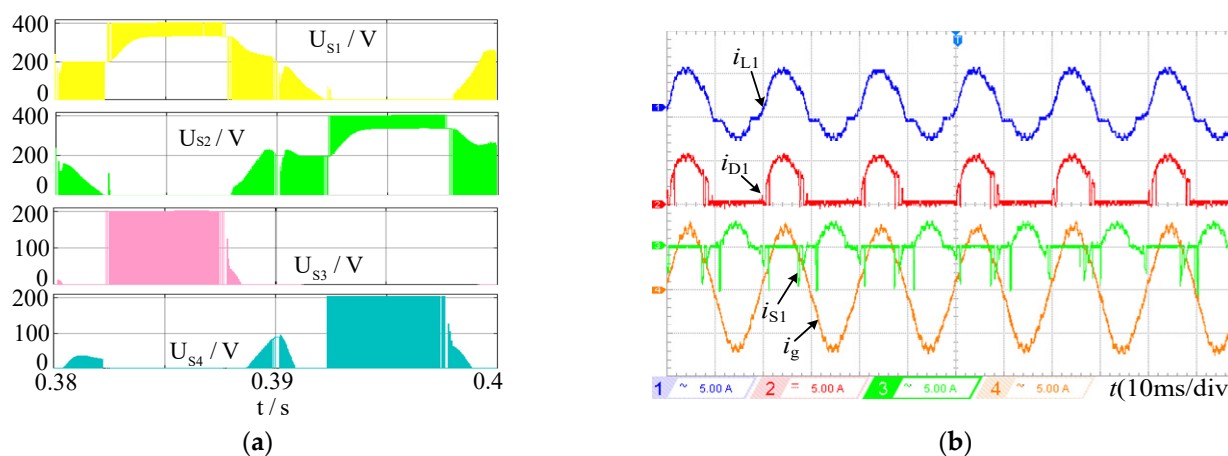

**Figure 17.** Voltage and current stress experimental waveforms. (**a**) The voltage stress waveform of switches $S_1$–$S_4$; (**b**) The inductor current $i_{L1}$, diode $D_1$ current $i_{D1}$, and the switch $S_1$ current $i_{S1}$.

Figure 18 shows the THD test results of PDBC-II at 1 kW rated power. As can be seen here, at the switching frequency of 20 kHz, the THD of PDBC-II is 3.2%, which meets the IEC 6100-3-2 standard.

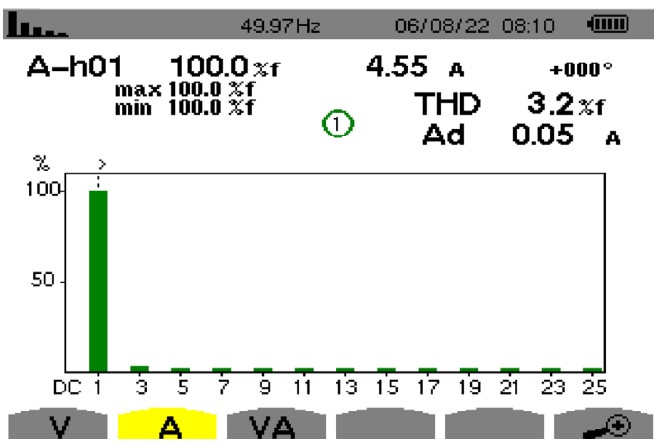

**Figure 18.** THD test results of PDBC-II.

## 6. Conclusions

In this paper, combining the three-level PTP circuit and the five-level NPC bridge arm, a family of PDBCs is proposed by transforming the NPC bridge arm. The proposed circuit reduces the volume of the filter inductor to increase the power density of the circuit and improves the working efficiency of the circuit. Through comparative analyses of voltage and current stress and loss, the following conclusions are drawn.

(1) The experimental results show that the PDBCs proposed in this paper have good input and output waveforms at low switching frequency. Therefore, the PDBCs can improve overall efficiency by reducing switching losses.

(2) The efficiency of the five PDBCs proposed in this paper is higher than that of a CFLC. The PDBC-I has the smallest loss and the highest efficiency, with a peak efficiency of 98.27%, and its overall performance is the best.

(3) Compared with the conventional three-level PTP circuit, the five PDBCs proposed in this paper have higher power density, and the voltage stress of some devices is reduced by half. In addition, the PDBC-II and PDBC-III have more devices with voltage stresses halved; thus, their cost is lower.

**Author Contributions:** Conceptualization, C.J.; Methodology, G.M.; Validation, Q.Z.; Resources, W.L.; Data curation, H.D.; Writing—original draft, Q.Z.; Writing—review & editing, J.X.; Supervision, H.M. All authors have read and agreed to the published version of the manuscript.

**Funding:** This research was funded by the Hubei Provincial Key Laboratory for Operation and Control of Cascaded Hydropower Station (China Three Gorges University), China grant number [2022KJX05].

**Data Availability Statement:** Not applicable.

**Conflicts of Interest:** The authors declare no conflict of interest.

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
