# Peer review of "A Family of Five-Level Pseudo-Totem Pole Dual Boost Converters"

_electronics, doi:10.3390/electronics12173722_

Round 1
Reviewer 1 Report
On page 2, line 16: "However, ...process". This short phrase is somehow ambiguous. Please, if possible, elaborate how this is true.
On page 2, from line 80 to page 3, to line 99: The topological derivation ¿has it been derived previously in other paper? ¿Is that new? Please, clarify.
On page 4 to 5, a suggestion is made to elaborate how the change of operating switch is made. I found no references in the text for the change of operation.
And finally, on the experimental validation: Why there is no data gathered to also validate the performance analysis?
Author Response
Reviewer 1
1、On page 2, line 16: "However, ...process". This short phrase is somehow ambiguous. Please, if possible, elaborate how this is true.
Response:Thanks to reviewer,the novel topology circuits are directly proposed in the most of papers without the process of topology derivation. This paper presents a family of five-level PTP dual boost converters (PDBC) based on the pseudo-totem pole PFC circuit. Modify this sentence to:However, there is no topology derivation process.
2、On page 2, from line 80 to page 3, to line 99: The topological derivation ¿has it been derived previously in other paper? ¿Is that new? Please, clarify.
Response:Thanks to reviewer, this derivation of the novel topology is proposed in this paper, the paper has been stated clearly in the second part ” 2. Topological Derivation and Operating Principle”. 2.1. Topological Derivation
3、On page 4 to 5, a suggestion is made to elaborate how the change of operating switch is made. I found no references in the text for the change of operation.
Response:Thanks to reviewer, this method, the replacement of switch components, is firstly proposed in this paper. In addition, the switch components are replaced according to the working mode of the PFC circuit.
4、And finally, on the experimental validation: Why there is no data gathered to also validate the performance analysis?
Response:Thanks to reviewer, this paper is mainly for the proposed PFC circuit topology, in the following experimental verification is for the feasibility of the proposed circuit experimental verification, the feasibility and validity of the circuit can be proved by the steady-state process and pulse of the circuit, and the feasibility of the proposed circuit derivation method can be verified by the experimental waveform.

Reviewer 2 Report
In order to improve the research work, a comment is made below
A comparison between theoretical and experimentally measured efficiency is necessary. The authors present only a plot of the theoretical efficiency obtained with the analysis.
A revision of the manuscript is desirable to correct possible grammar and writing errors.
Author Response
Reviewer 2
1、In order to improve the research work, a comment is made below
A comparison between theoretical and experimentally measured efficiency is necessary. The authors present only a plot of the theoretical efficiency obtained with the analysis.
Response:Thanks to reviewer, this paper is mainly for the proposed PFC circuit topology, in the following experimental verification is for the feasibility of the proposed circuit experimental verification, the feasibility and validity of the circuit can be proved by the steady-state process and pulse of the circuit, and the feasibility of the proposed circuit derivation method can be verified by the experimental waveform.
The feasibility of the proposed circuit is verified by experiments. The feasibility and validity of the circuit can be proved by steady-state process and pulse The experimental waveforms can verify the feasibility of the proposed circuit derivation method.
2、A revision of the manuscript is desirable to correct possible grammar and writing errors.
Response:Accept the reviewer’s suggestion, read the whole paper and revise the paper.

Reviewer 3 Report
This paper presents a family of five-level pseudo-totem pole dual boost Converters.
1. In the abstract the authors stated: "Compared with conventional three-level PTP converters, PDBCs have higher power density, higher efficiency, and low voltage and current stress.". Anyway, there is no evidence in the paper. Please justify better.
2. In addition to the current stress analysis in Section 4.2, also the voltage waveforms across the components should be presented.
3. The loss analysis shown in Section 4.3 should be performed at different loads. The measurements are performed at Io=Po/Vo=2.5 A, which is a low current. I suggest evaluating the power losses also at higher current ratings.
4. Equations (20) is a very simplistic approximation. In practice, the rise and fall times are a function of the current and voltage on the MOSFET. It is advisable to use more accurate models or at least to mention the fact that the phenomenon is complex but modelable with models available in the literature. Examples:
- E. Locorotondo et al., "Analytical Model of Power MOSFET Switching Losses due to Parasitic Components," 2019 IEEE 5th International forum on Research and Technology for Society and Industry (RTSI), Florence, Italy, 2019, pp. 331-336, doi: 10.1109/RTSI.2019.8895562.
- C. Entzminger, W. Qiao, L. Qu and J. L. Hudgins, "A High-Accuracy, Low-Order Thermal Model of SiC MOSFET Power Modules Extracted from Finite Element Analysis via Model Order Reduction," 2019 IEEE Energy Conversion Congress and Exposition (ECCE), Baltimore, MD, USA, 2019, pp. 4950-4954, doi: 10.1109/ECCE.2019.8912839.
4. Since the aim is to work with high power density, why f=20kHz has been used? The adoption of higher switching frequency (and the effects on the power losses) should be discussed.
5. Which are the parasitics of passive components?
6. What is the THD of the source current ig? Is the one in Fig. 18? Please clarify.
Moderate editing of English language required
Author Response
Reviewer 3
- In the abstract the authors stated: "Compared with conventional three-level PTP converters, PDBCs have higher power density, higher efficiency, and low voltage and current stress.". Anyway, there is no evidence in the paper. Please justify better.
Response:Thanks to reviewer , this sentence really is not suitable here, and it has little to do with the topic of the paper, So we will delete it.
This paper mainly aims at the circuit topology, and then the feasibility of the proposed circuit is verified by the experiment. The feasibility and validity of the circuit can be proved by the circuit steady-state process and pulse.
- In addition to the current stress analysis in Section 4.2, also the voltage waveforms across the components should be presented.
Response:Thanks to reviewer, the current stress analysis here is corresponding to the current calculation theory, for the voltage stress analysis as follows: Table 2; and for the device stress in the part 5 experimental section, the voltage waveform is given, as shown in the figure 15, the proposed 5-level circuit is further described.
- The loss analysis shown in Section 4.3 should be performed at different loads. The measurements are performed at Io=Po/Vo=2.5 A, which is a low current. I suggest evaluating the power losses also at higher current ratings.
Response:In this paper, the PFC circuit is designed in 1KW, the DC current is 2.5 A, and the efficiency analysis is in 1kW experimental platform. This paper is mainly for the proposed PFC circuit topology, and the topology feasibility is analyzed with rated power 1kw.
- Equations (20) is a very simplistic approximation. In practice, the rise and fall times are a function of the current and voltage on the MOSFET. It is advisable to use more accurate models or at least to mention the fact that the phenomenon is complex but modelable with models available in the literature. Examples:
Response:Thanks to the reviewer's suggestions, efficiency analysis part of this paper on circuit topology, this paper uses simplified engineering approach for efficiency analysis in loss analysis.
- Since the aim is to work with high power density, why f=20kHz has been used? The adoption of higher switching frequency (and the effects on the power losses) should be discussed.
Response:Thanks to the reviewer's suggestions,This question is indeed a good question, but the core point of this paper is to deduce the topology and build the experimental circuit for experimental verification; the questions raised by experts are not the ones discussed in this paper, thank you.
- Which are the parasitics of passive components?
Response:Thanks to the reviewer's suggestions, the parasitic parameters are non-essential elements existing in electronic components or circuits, which will adversely affect the normal operation of the components or circuits. It usually displays as the problems of consuming power, limiting signal transmission rate or affecting signal quality. It is necessary to minimize or avoid the generation of parasitic parameters. Parasitic parameters are elements such as capacitors, inductors and resistors that are not necessary in electronic component or circuits. These elements are usually caused by the physical properties of wires, circuit boards, and other circuit elements. The parasitic parameters damage the circuit performance, such as reducing frequency response, increasing noise, reducing signal amplitude and so on.
Therefore, circuit designers need to fully consider the impact of parasitic parameters, and take measures to reduce its impact.
The main of this paper is to deduce the topology and build the experimental circuit to verify it. The parasitic parameters of passive devices are not discussed too much in this paper.
- What is the THD of the source current ig? Is the one in Fig. 18? Please clarify.
Response:THD is Total Harmonic Distortion;

Reviewer 4 Report
The submitted manuscript presents a novel five-level Pseudo-Totem Pole Dual Boost Converter that opimizes the three-level Pseudo-Totem Pole converter in terms of volume, switching loss and reliability. The manuscript is overall well written and of interest to the power converters community. Only a few comments are due. The manuscript needs revision on the English quality perspective since some errors can be spotted (for instance "is send", "is multiply", "which using" and others). In Section 2.1 the numbering of the bridge arms A1, A2, etc... in the text (between line 80 and 85) is wrong. In the last part of Section 2.1, Authors claim that "It is worth mentioning that the three power flow converters proposed in this paper can all achieve bidirectional power flow after a slight circuit modification." Can you provide more insight about what kind of circuit modification is needed to achieve bidirectional power flow? In Section 3.2, why the control is constructed using simple PI controllers? The Reviewer would suggest to mention that more complex nonlinear control strategies could be designed for the converter control. For instance, in [Stability and Control for Buck–Boost Converter for Aeronautic Power Management. Energies 2023, 16, 988], higher order sliding mode control algorithms are designed for power converter control. Moreover, which strategy did you use to tune the controllers parameters?
A few typos can be spotted in the manuscript. Authors are required to revise the writing.
Author Response
Reviewer 4
1、The submitted manuscript presents a novel five-level Pseudo-Totem Pole Dual Boost Converter that opimizes the three-level Pseudo-Totem Pole converter in terms of volume, switching loss and reliability. The manuscript is overall well written and of interest to the power converters community. Only a few comments are due. The manuscript needs revision on the English quality perspective since some errors can be spotted (for instance "is send", "is multiply", "which using" and others).
Response:Thanks to the reviewer, we will check and correct.
2、In Section 2.1 the numbering of the bridge arms A1, A2, etc... in the text (between line 80 and 85) is wrong. In the last part of Section 2.1, Authors claim that "It is worth mentioning that the three power flow converters proposed in this paper can all achieve bidirectional power flow after a slight circuit modification." Can you provide more insight about what kind of circuit modification is needed to achieve bidirectional power flow?
Response:Thanks to reviewer, this paper is mainly for the proposed PFC circuit topology, in the following experimental verification is for the feasibility of the proposed circuit experimental verification, the feasibility and validity of the circuit can be proved by the steady-state process and pulse of the circuit, and the feasibility of the proposed circuit derivation method can be verified by the experimental waveform.
3、In Section 3.2, why the control is constructed using simple PI controllers? The Reviewer would suggest to mention that more complex nonlinear control strategies could be designed for the converter control. For instance, in [Stability and Control for Buck–Boost Converter for Aeronautic Power Management. Energies 2023, 16, 988], higher order sliding mode control algorithms are designed for power converter control. Moreover, which strategy did you use to tune the controllers parameters?
A few typos can be spotted in the manuscript. Authors are required to revise the writing.
Response:Thanks to the reviewer, we will check and correct

Round 2
Reviewer 2 Report
As the authors mention, the experimental results validate the proposed configuration. In Fig. 13, a theoretical graph of the efficiency of the proposed converter is shown, which must be validated experimentally.
Author Response
As the authors mention, the experimental results validate the proposed configuration. In Fig. 13, a theoretical graph of the efficiency of the proposed converter is shown, which must be validated experimentally.
Response:Thanks . The paper gives the calculation process of efficiency, the Fig.13 shows the actual efficiency calculation curve of the 1KW experimental prototype, the efficiency analysis is the result of the analysis under the 1kW experimental PFC circuit. The Fig. 13 is the result of the experimental efficiency analysis.
Reviewer 3 Report
In the reviewer's opinion, considering the new material and modifications carried out by the paper authors, this new version of the manuscript can be accepted for publicantion in the Journal.
Author Response
In the reviewer's opinion, considering the new material and modifications carried out by the paper authors, this new version of the manuscript can be accepted for publicantion in the Journal.
Thank you very much
Round 3
Reviewer 2 Report
Plots shown in Fig. 13 between old version and new version are the same. The authors comment that an experimental plot of power efficiency was added in the new version, but no additional plot was added.
Author Response
Response:Thanks to the reviewer's suggestion. The PDBC-II circuit is built the experimental prototype, and the experimental efficiency analysis of the PDBC-II circuit was supplemented, Fig. 13(b)shows the experimental results of the PDBC-II circuit. Based on the room temperature, the efficiency analysis is expressed in the Fig. 13(b). There are actual circuits exist auxiliary circuit (drive circuit, sampling circuit,et.al), so the actual loss is higher than the theoretical analysis result. For theoretical calculation and practical experimental results, the trend of overall efficiency distribution is consistent.
